# Effect of High-Pressure Processing on the Packaging Properties of Biopolymer-Based Films: A Review

**DOI:** 10.3390/polym14153009

**Published:** 2022-07-25

**Authors:** Monjurul Hoque, Ciara McDonagh, Brijesh K. Tiwari, Joseph P. Kerry, Shivani Pathania

**Affiliations:** 1Food Industry Development Department, Teagasc Food Research Centre, Ashtown, D15 KN3K Dublin, Ireland; monjurul.hoque@teagasc.ie (M.H.); ciara.mcdonagh@teagasc.ie (C.M.); 2School of Food and Nutritional Sciences, University College Cork, T12 R229 Cork, Ireland; joe.kerry@ucc.ie; 3Food Chemistry and Technology Department, Teagasc Food Research Centre, Ashtown, D15 KN3K Dublin, Ireland; brijesh.tiwari@teagasc.ie

**Keywords:** high-pressure processing, film-forming solution, biopolymer-based packaging, morphological properties, mechanical properties, thermal properties, barrier properties, migration potential

## Abstract

Suitable packaging material in combination with high-pressure processing (HPP) can retain nutritional and organoleptic qualities besides extending the product’s shelf life of food products. However, the selection of appropriate packaging materials suitable for HPP is tremendously important because harsh environments like high pressure and high temperature during the processing can result in deviation in the visual and functional properties of the packaging materials. Traditionally, fossil-based plastic packaging is preferred for the HPP of food products, but these materials are of serious concern to the environment. Therefore, bio-based packaging systems are proposed to be a promising alternative to fossil-based plastic packaging. Some studies have scrutinized the impact of HPP on the functional properties of biopolymer-based packaging materials. This review summarizes the HPP application on biopolymer-based film-forming solutions and pre-formed biopolymer-based films. The impact of HPP on the key packaging properties such as structural, mechanical, thermal, and barrier properties in addition to the migration of additives from the packaging material into food products were systemically analyzed. HPP can be applied either to the film-forming solution or preformed packages. Structural, mechanical, hydrophobic, barrier, and thermal characteristics of the films are enhanced when the film-forming solution is exposed to HPP overcoming the shortcomings of the native biopolymers-based film. Also, biopolymer-based packaging mostly PLA based when exposed to HPP at low temperature showed no significant deviation in packaging properties indicating the suitability of their applications. HPP may induce the migration of packaging additives and thus should be thoroughly studied. Overall, HPP can be one way to enhance the properties of biopolymer-based films and can also be used for packaging food materials intended for HPP.

## 1. Introduction

Currently, the consumer trend is towards a growing demand for minimally processed food with improved food safety, nutritional value, freshness, and natural flavors. In order to meet these demands, food industries have been using different processing technologies allowing reduced additives without compromising the sensory and nutritional qualities of the food material. Among different technologies, high-pressure processing (HPP) is one of the emerging noble technologies in the food industry that produce clean label foods free from chemical additives and causes minimal product quality loss along with extending shelf-life by inactivating microbes and enzymes [1,2,3,4]. HPP technology accomplishes these by applying high hydrostatic pressures ranging from 100 -1000 MPa to food products. The United States Department of Agriculture Food Safety and Inspection Service (USDA-FSIS) has approved HPP to be applied on both raw and ready-to-eat products [5]. Globally, in 2015, more than 500,000 tons of food products were produced using HPP technology mostly in the area of meat (27%), dairy and egg products (20%), aquatic (12 %), fruits and vegetables (27%), and beverage (14%). In 2015, the global market for HPP food products was USD 9.8 billion and is expected to be USD 54.77 billion by 2025 [1,6].

Figure 1 shows that HPP comprises initial heating (if required) of the hermetically sealed food products, followed by the application of adiabatic pressure using a pressure transmitting medium. The commonly used transmitting medium includes water, food-grade glycol-water solutions, silicon oil, sodium benzoate solution, ethanol, and castor oil [4,7]. As the process is adiabatic compression, according to the principle of compression heating, a monotonous increase in the initial temperature is determined and with the rise in pressure by every 100 MPa, the temperature is enhanced by 3 °C. However, this increment is dependent on the pressure transmitting medium and properties of the food products. Isostatic principle and Pascal’s principle govern the uniform distribution of pressure on the food materials in the sealed vessel. As per the isostatic principle, upon application of pressure to a liquid medium in a closed chamber, equal pressure is experienced by the object placed at any point within the chamber regardless of shape and size. According to Pascal’s principle, any phenomenon accompanied by a decrease in volume is enhanced by pressure. If the pressure changes, the equilibrium shifts in a direction that tends to lessen the change in the corresponding intensive variable (volume) [2,3,4].

Commercially, HPP treatment is carried out in batch, semi-continuous, or continuous mode. Liquid products can be treated in a semi-continuous or continuous process whereas solid products are only treated in batch mode [8]. It has been documented that almost 90% of commercial HPP food products are processed in batch systems, where flexible or partially rigid materials are used as the packaging material prior to processing [9,10]. The selected packaging system must be adequate to withstand volume changes (compression up to 15%) and be able to return to its original shape (decompression) without leaching undesirable packaging chemicals into the product [11]. Also, the packaging materials should be able to withstand the operating conditions of HPP and should have sufficient mechanical, heat sealing integrity, and barrier properties to avoid damage to the product during the processing and distribution in the market [1,12].

In the food industry, fossil-based plastic is the ultimate choice for HPP of packaged food materials due to its excellent thermo-mechanical properties with high strength and flexibility, low cost, lightweight, shape versatility, high performance, and easy transportation. Among, others, polyethylene terephthalate (PET), polyethylene (PE), polypropylene (PP), polyamide (PA), and ethylene-vinyl alcohol (EVOH) are used in food industries [13,14,15]. López-Rubio, Lagarón [16] reported that ethylene-vinyl alcohol-based food packaging materials subjected to 400 and 800 MPa showed no detrimental effect on the barrier and morphological properties. Marangoni Junior, Alves [17] applied 600 MPa at different time-temperature combinations and observed no significant change in water barrier and light barrier properties of multilayer films such as LDPE/PA/LDPE, LDPE/EVOH/LDPE, PET/LDPE/PA/EVOH/PA/LDPE, and PET/Al/PA/PP. Although conventional plastics demonstrate excellent suitability for HPP, these are non-biodegradable and non-renewable. The large application has raised plastic production and its worldwide production was 368 million metric tons in 2019 and has been continuously growing at 4% every year [18,19]. The critical point of high plastic production is the manufacturing of single-use plastics (almost 50%) that cause the issue of plastic pollution and its waste management. Only 9% of all the plastics get recycled and the rest ends in the land fields and water bodies. This high production of plastics is not only a serious threat to individuals, or communities but to the whole ecosystem, especially the marine ecosystem [18,20].

Biopolymer-based compostable packaging is one of the alternatives to plastic packaging. Biopolymers like polysaccharides and proteins from plant or animal origins are being explored to develop packaging materials. Previous studies showed that biopolymer-based films have good film-forming capacity but this packaging system exerts certain shortcomings like low mechanical strength, high hydrophilicity, and poor barrier property as compared to synthetic plastics [21,22,23,24]. These limitations can be overcome by the application of HPP. As stated by Pascal’s principle, the application of high pressure reduces volume, and thus biopolymer-based packaging materials when subjected to HPP produce a compact network microstructure that enhances mechanical strength and barrier properties [1,25]. The HPP also promotes modification of macromolecular arrangements like starch gelatinization, and protein denaturation and increases interaction between different components [26,27,28,29]. Therefore HPP has been proposed as an effective technique to overcome the shortcomings of biopolymer-based materials and several studies have reported forming much denser and more uniform packaging with desired properties [23,30,31].

Until now, HPP has been found to be effective in the modification of biopolymers like polysaccharides [31], proteins [30], and bioplastics like PLA [32]. The HPP technique can be used to modify the properties of biopolymer in two forms as shown in Figure 2. HPP can be applied to the film-forming solution (FFS) before drying (Figure 2a) or can be applied to the pre-formed packaging films (Figure 2b). The sequence of application and processing conditions (pressure, time, and temperature) results in different properties. In this review, the authors systematically analyzed the performance change in the morphological, mechanical, barrier properties, and thermal properties of the biopolymer-based packaging material when subjected to HPP treatment (with different pressure levels, time, and temperature) before and after film formation.

## 2. Impact of HPP When Applied to the Film-Forming Solution (FFS)

Bio-based polymers such as polysaccharides, proteins, or their derivatives show good film-forming capacity and are believed to be futuristic replacements for synthetic plastics. Although these polymers form films, due to certain limitations in their pristine forms, some modifications are reported to enhance their packaging properties. Different physical and chemical modifications of biopolymers are being carried out. HPP is a novel technique that utilizes high pressure of more than 100 MPa and is primarily used for the preservation of food products in food processing industries. Recently, it has also been applied in the physical modification of biopolymer-based films [32,33]. For the modification of the packaging properties of the film, high pressure can either be applied to the film-forming solution or the prepared film. HPP when efficiently applied can induce macromolecular changes like protein denaturation or starch gelatinization that may influence the packaging properties of the biopolymer-based films [1,29,34,35]. Different biopolymers interact differently when subjected to pressure and this section elaborates on the impact of HPP on the different properties of the films when applied to the film-forming solution (FFS).

### 2.1. Surface Attributes

According to the literature, the surface roughness of the biopolymer-based film subjected to HPP largely depends on the level of pressure applied and the composition of the films. It is observed that FFS, when treated with HPP, mostly results in uniform and smooth surfaces. Wei, Zhang [30] studied the microstructural attributes of nisin-incorporated soy protein isolate-based films over a range of 100–500 MPa. Control films exhibited a compact and uniform surface with a few blocks of polymers embedded in the surface but as HPP was employed, a reduction in size and amount of embedded polymers was observed, resulting in a smooth and homogenous surface. The cross-sectional microstructure confirmed the finer and smoother structure of the film and is shown in Figure 3a. This increase in fineness was attributed to the uniform polymer dispersion formed from a lower viscous suspension, achieved through HPP treatment [30]. 

Another microstructural study on HPP treatment (200 and 600 MPa) of polyvinyl alcohol and chitosan (PVA/CHI) loaded with nano-TiO_2_ composite films demonstrated that HPP can diminish the transition layer between PVA and CHI (Figure 3b). Cross-sectional studies of PVA/CHI/TiO_2_ films revealed that a higher pressure of 600 MPa resulted in a finer and smoother surface as compared to the films processed at 400 MPa which is attributed to the formation of new hydrogen bonds between amino groups of PVA and CHI at 600 MPa pressure [36]. The increase in smoothness and fineness of the film surface post HPP treatment was also reported in buckwheat and tapioca starch [31] films. However, Chi, Xue [32] reported a contrasting effect, that is, increased surface roughness in PLA/Ag NPs-based films treated at 200 and 400 MPa (please refer to Figure 3c). Such difference in surface properties post-HPP-treatment could be owed to the inherent polymer and the nanomaterial properties as well their interaction upon HPP treatment.

### 2.2. Mechanical Attributes

Tensile strength (TS) and elongation at break (EAB) are critical mechanical properties associated with the performance of automatic packaging lines and also maintain food quality and integrity during handling, processing, shipping, and storage [4,18]. HPP of FFS can cause the volume-driven transition, promotes cross-linking, and induces phenomena such as protein denaturation or starch gelatinization that can affect the mechanical properties of the films [29,30,31,34,37]. The impact of HPP on the mechanical properties of the film depends on several factors like the level of pressure used, time of exposure, types of bonds in the matrix, and the presence of cross-linkers. The mechanical results of HPP on the preparation of biopolymer-based films are listed in Table 1.

In the case of starch, HPP treatment induces volume-driven transition like the transformation of starch crystalline structure from A- to B-type. Films prepared from HPP-treated FFS possess higher moisture content that helps in stabilizing the scattered amylopectin structure of A-type starches via van der Waals forces, resulting in the rearrangement of the helix structure leading to the formation of compact B-type structure [31,38,39,40]. In this regard, Kim, Yang [31] stated that the TS value of HP-treated buckwheat starch (BS) was higher (18.29 ± 1.05 MPa) as compared to the untreated film (13.61 ± 1.06 MPa); similarly, HP-treated tapioca starch (TPS) films showed higher TS of 26.92 ± 0.43 MPa as compared to untreated film 24.67 ± 1.03 MPa. It was also found that increment in the TS values of BS was higher than that of TPS and this was ascribed to the fact that BS has A-type starch which is more sensitive to HPP compared to C-type starch present in the TPS starch [31]. Also, the application of HPP (600 MPa at 20 °C for 20 min) increased the EAB of BS film from 5.65 ± 0.23% to 7.92 ± 0.58%, and the EAB of TPS film increased from 5.04 ± 0.56% to 5.71 ± 0.20%. This enhancement in the extensibility of the film could be due to the plasticizing effect of the higher moisture present in the HPP-treated film as compared to the untreated film [31].

In the case of protein-based film, HPP is an effective factor that can induce the formation of hydrogen bonds, and disulfide bonds or may increase hydrophobic interaction between the components of the film, resulting in the formation of a more compact structure that leads to the considerable enhancement in the mechanical properties. Molinaro, Cruz-Romero [41] reported that with an application of high pressure (600 MPa at 20.5 °C for 30 min) to the FSS from pigskin gelation-(PSG), the TS value significantly increased from 25.7 ± 2.2 to 28.7 ± 2.5 MPa; and EAB increased from 8.6 ± 0.6 to 10.1 ± 1.5%. This enhancement could be due to the increase in the stiffness in the gelatin matrix due to the formation of H-bonds. It was also observed that the different pressure level applied to the FFS influences the mechanical properties. As in the case of soy protein isolate (SPI) based FFS, application of HP from 100 to 500 MPa at 20 °C for 10 min progressively increased hydrophobic interactions between the SPI along with the formation of disulfide bonds resulting in the enhancement in the TS values but decreased EAB [30]. Similarly, an increase in the pressure level from 200 to 400 and then 600 MPa in amaranth protein-based FFS, increased the TS of the film by 26, 101, and 165%, respectively, as compared to untreated film, but the EAB was unaltered [37]. 

HPP processing of FFS, containing nanoparticles also significantly influences the mechanical properties of the film. Lian, Zhang [36] showed that polyvinyl alcohol-chitosan-TiO_2_ (0.10%) FFS when subjected to 200, 400, and 600 MPa increased the TS from 8.24 ± 0.27 MPa to 13.67 ± 0.41, 13.98 ± 0.33, and 17.15 ± 0.97 Mpa, respectively, and EAB also increased from 64.82 ± 1.10% to 68.48 ± 1.66, 68.12 ± 1.94, and 67.92 ± 2.73%, respectively, although the pressure level showed no significant impact. Similarly, TS of PLA-Ag-3% or PLA-Ag-5% when subjected to 200 and 400 MPa increased the TS but decreased the EAB as shown in Table 1 [32]. Such a trend in mechanical properties where TS increased and EAB decreased with the increased pressure level (0, 200, and 400) also was reported for PLA film loaded with ZnO NPs at 0, 2.5, 5.0, and 10.0 % of PLA [42]. Such variation in the mechanical properties of the nanocomposite film exposed to HPP can be ascribed to the enhancement in the crystallinity, development of ordered molecular chain arrangement, and reduction in porosity as well as the increase in the intermolecular interaction resulting in the restricted chain mobility [32,42].

### 2.3. Water Solubility (WS)

Solubility in water is one of the key features of biopolymer-based film. In food packaging applications, packaging material should be water-insoluble to protect the integrity of the packaged product, prevent moisture transport, and enhance shelf-life [21,22]. The application of HPP is expected to reduce the WS of the biopolymer-based films. Table 1 summarizes the WS results of HPP on the preparation of biopolymer-based films. In one of the studies, Kim, Yang [31] reported that application of 600 MPa at 20 °C for 20 min to the FFS buckwheat starch reduced the WS of the film from 19.85 ± 0.33 to 11.67 ± 0.69%; and tapioca starch from 28.53 ± 0.68 to 17.53 ± 0.51%. This decrease in WS could be due to the formation of a compact helix structure in the starch-based matrix through hydrogen bonding that might have reduced the hydroxyl site for the interaction with water molecules and thus reduced water solubility. Similarly, Kim, Choi [43] reported that corn starch exposed to 400 MPa, at 25 °C, for 15 min showed lower water solubility compared to the conventional thermal processing of starch. Such decrease in WS of the protein-based film was reported for amaranth protein isolate films and it decreased progressively with the increase in the intensity of the pressure. It was reported that WS of the untreated film 79.9 ± 2.1% decreased to 56.4 ± 5.5, 46.1 ± 0.5, and 46.1 ± 2.5% for the film prepared from FSS subjected to 200, 400, and 600 MPa for 5 min, respectively. This decrease in WS could be due to the higher crosslinking of the film developed from the protein, unfolded by HPP. Also, the unfolded protein might have facilitated different interactions resulting in a stable and compact protein matrix with higher surface hydrophobicity that might have retarded the passage of water molecules through it and retained the film structure when in contact with water [37].

### 2.4. Barrier Property

Polysaccharides are one of the primary candidates for the preparation of films for food packaging applications. It has been observed that the application of HPP has a significant impact on the film-forming properties of polysaccharides like chitosan, carrageenan, starch, or their derivatives. Table 1 summarizes the barrier properties of HPP in the preparation of biopolymer-based films. HPP influence the gelatinization of starch and thus their film characteristics. Kim, Yang [31] prepared buckwheat starch (BS) and tapioca starch (TPS) film. The BS film prepared by treating the FFS at 600 MPa for 20 min showed significantly lower WVP (2.10 × 10^−9^ g/m s Pa) compared to film prepared by heating at 90 °C for 20 min, where the WVP was 3.10 × 10^−9^ g/m s Pa. This decrease in WVP may be attributed to the creation of the denser matrix with the application of HPP, which might have restricted the water movement. However, the impact of HPP (600 MPa) on the WVP of the TPS film was less evident as compared to the HP-treated BS film. This variation in WVP upon application of HPP at the same condition may be due to the structural differences between starches that might have influenced the extent of gelatinization. The degree of gelatinization is not only dependent on the starch type but on water concentration and processing parameters (pressure level, holding time, and temperature). Also, among the different types of starch (A-, B-, and C-), type A- is most sensitive to HPP influencing the film-forming properties [1,44]. BS contains A-type starch whereas TPS has C-type starch and thus BS showed lower WVP when subjected to HPP [31]. Similarly, Niu, Chen [45] reported that the application of HPP (100 to 500 MPa) to the chitosan FFS for 15 min decreased the WVP and OP significantly compared to the untreated film. This enhancement in the barrier property could be due to the re-formation of a stable and compact structure of chitosan when exposed to HP treatment. It is expected that HPP can cleavage hydrogen and hydrophobic bonds exposing polar groups and facilitating their rebinding to form a compact and stable structure resulting in the enhancement in the barrier properties [45]. 

Another class of biodegradable and compostable thermoplastic polyester already commercialized as a compostable packaging material is poly-lactic acid (PLA) produced from L- and D- lactic acid obtained by bacterial fermentation of starch. However, the poor barrier property of pristine PLA is a limitation in its commercial application. Therefore, several attempts are being made to enhance its barrier property. Among others, physical modification of PLA by incorporating nanoparticles and application of HP has been proven to be outstanding. Chi, Xue [32] applied a high pressure of 200 and 400 MPa to the PLA-Ag nanocomposite FFS and investigated the properties of the film. It was reported that the application of HP reduced the water vapor permeability (WVP) of the nanocomposite film by 51.5 and 44.3% upon the addition of 5.0% Ag NPs and application of 200 and 400 MPa pressure, respectively. HPP might have increased the interaction between PLA and the Ag nanoparticles via hydrogen bonding and van der Waals forces and enhanced the compactness of the biopolymer network resulting in reduced WVP. Also, the incorporation of Ag NPs into the PLA matrix and simultaneous application of HP treatment improved the crystallinity of the composite film and increased the compactness contributing to the reduction of WVP [32]. Similarly, PLA/ZnO NPs FFS treated with HP maintained at 200 and 400 MPa showed that WVP and oxygen permeability (OP) decreased by almost 45 and 38%, respectively, with the addition of 50 g/kg ZnO NPs and application of 400 MPa. The decrease in the WVP and OP can be attributed to the increase in the crystallinity of the film leading to higher tortuosity in the transportation path and the interaction between PLA and hydroxyl groups on the surface of ZnO NPs resulting in the compactness of the film [42]. Overall, it can be seen that the incorporation of NPs at optimal concentration and simultaneous HP treatment can increase the crystallinity and interaction between the components of the film matrix resulting in the improvement of the barrier properties. 

Similar to the polysaccharide, protein when exposed to HPP induce modification by influencing the gel formation that is mainly associated with protein denaturation, dissociation-association, and aggregation. In the HPP environment, disruption of the quaternary and tertiary structure of globular proteins occurs with little influence on the secondary structure. However, functional groups present in the unfolded portion of the proteins are exposed to each other and form a more compact structure with the formation of new hydrogen bonds due to the negative activation volume and enthalpy [1,41]. Protein-based FFS when treated with HP shows improvement in the barrier properties of the film. Wei, Zhang [30] treated an FFS of soy protein isolate (SPI)/nisin with HP (100–500 MPa) at 20 ± 0.5 °C for 10 min. It was reported that HP treatment significantly reduced the WVP by more than 50% compared to the control film due to the formation of the compact and uniform matrix under the influence of HPP. Also, the HP treatment unfolds the SPI protein exposing the more hydrophobic groups at the water-air interface resulting in the enhancement of hydrophobicity of the film and thus increasing the water barrier property [30]. Similarly, Molinaro, Cruz-Romero [41] treated pigskin-derived gelatin-based FFS at different levels of HPP, holding time, and temperature. At an optimal condition of 600 MPa, for 30 min at 20.45 °C, OTR and WVTR significantly decreased from 2.90 to 1.83 mL m^−2^ day^−1^ and 65.56 ± 1.2 to 63.47 ± 0.9 g/(day m^2^), respectively. This enhancement in the barrier properties could be due to the formation of a large number of hydrogen bonds and stable short critical helix leading to the development of compact structure under the influence of the optimal HPP conditions [41]. Similarly, Koehler, Kieffer [46] studied the effect of HPP on wheat gluten III, and Condés, Añón [37] studied amaranth protein and showed significant enhancement in the barrier properties.

### 2.5. Thermal Properties

Differential scanning calorimetry (DSC) is one of the most widely used approaches to determine the variations in heating/cooling properties as well as thermodynamic or thermophysical characteristics of biopolymers. The important thermal parameters analyzed by DSC methods include onset temperature-T_o_, glass transition temperature-T_g_, melting temperature-T_m_, crystallization temperature-T_c_, and enthalpy-ΔH. Assessing these parameters can determine structural and thermal changes induced by HPP. These parameters are also significant to determine the processing conditions of biopolymers and their applications in packaging, like heat sealing [18,47]. Table 1 summarizes the thermal properties of HPP in the preparation of biopolymer-based films.

The HPP has diverse effects on the thermal profile of biopolymer-based films and in some cases, it is observed that the thermal stability is enhanced. Kim, Yang [31] investigated the HPP on the thermal behavior of buckwheat (BS) and tapioca starch (TPS) films. The result showed that application of 600 MPa at 20 °C for 20 min to BS film-forming solution (FFS), enhanced the T_o_ from 70.52 to 76.16 °C, T_m_ from 112.75 to 120.64 °C, and ΔH increased from 78.64 to 79.30 J/g. Similarly, HP-treated TPS film demonstrated that T_o_ increase from 70.92 to 84.32 °C, and ΔH increased from 56.92 to 78.40 J/g, but T_m_ slightly decreased from 124.62 to 122.07 °C. The enhancements in these parameters indicate that HPP might have enhanced the interactions between the film components and thus high thermal energy is required to dissociate these interactions. Also, the higher ΔH for both BS and TPS film could be due to the formation of stable amylopectin double-helix structure and the existence of a relatively higher percentage of crystalline structure in the HPP films as compared to the untreated film [39,48]. Similar thermal stability was observed in protein-based film prepared by HPP of pigskin-derived gelatin [41]. It was reported that HPP had no significant impact on T_g_ of the film but T_m_ increased from 131.5 ± 0.7 to 138.2 ± 0.5 °C suggesting the formation of hydrogen bonds (between NH and C=O of glycine and proline, respectively) during HPP of the gelatin. The significant decrease in ΔH values from 46.4 ± 0.8 to 36.5 ± 3.3 J/g could be attributed to the elastic or conformational changes in proteins and depression in the crystallinity within the protein matrix due to the mechanical forces experienced during HPP [41,49].

The PLA-based nanocomposite film prepared by HPP of FFS demonstrated thermal stability. PLA-AgNPs-5% FFS when subjected to 400 MPa at 25 °C for 15 min showed a significant increment in T_g_ from 50.1 ± 0.2 to 51.9 ± 0.2 °C, which could be due to the application of HP that might have restricted the PLA chain mobility and increased the T_g_ [32]. Such increment in T_g_ was also observed for PLA/ZnO nanocomposite film prepared from FFS subjected to 400 MPa [42]. It was also observed that the application of HPP in FFS of PLA nanocomposite film, increased the T_c_, as evident in PLA-Ag-5% nanocomposite film, where T_c_ increased from 110.4 ± 0.4 to 112.9 ± 0.5 °C, also in PLA-ZnO-5% nanocomposite film, T_c_ increased from 95.9 ± 0.30 to 100.9 ± 0.70 °C, where FFS for both the films were subjected to 400 MPa. This enhancement in thermal stability could be attributed to the increase in the crystallinity due to the application of HPP in FFS, as it was found that percentage crystallinity increased from 15.8 ± 0.6% to 23.9 ± 0.4% for PLA-Ag-5% and 14.9 ± 0.74 to 20.4 ± 0.42% for PLA-ZnO-5% [32,42]. Overall, it was observed that the application of HPP to the FFS can enhance the thermal properties of biopolymer-based films.

**Table 1 polymers-14-03009-t001:** Effect of HPP on the film forming solution (FFS).

Film Matrix	Processing Conditions	Water Solubility (WS)	Barrier Property (WVP/OP)	Mechanical Property		Thermal Properties	R
				TS	EAB		
Buckwheat starch (BS)	600 MPa at 20 °C for 20 min	WS of the thermally processed BS film was 19.85 ± 0.33% significantly decreased to 11.67 ± 0.69% upon application of 600 MPa	WVP of the thermally processed BS film 3.10 × 10^−9^ g/m s Pa significantly decreased to 2.10 × 10^−9^ g/m s Pa, upon application of 600 MPa	TS of the thermally processed BS film 13.61 ± 1.06 MPa significantly increased to 18.29 ± 1.05 MPa upon application of 600 MPa	EAB of the thermally processed BS film 5.65 ± 0.23% significantly increased to 7.92 ± 0.58% upon application of 600 MPa	T_o_, T_m_, and ΔH of thermally processed BS film 70.52 °C, 112.75 °C, and 78.64 J/g increased to 76.16 °C, 120.64 °C, and 79.30 J/g, respectively; upon application of 600 MPa	[31]
Tapioca-starch (TPS)	600 MPa at 20 °C for 20 min	WS of the thermally processed TPS film 28.53 ± 0.68% significantly decreased to 17.53 ± 0.51% upon application of 600 MPa	No significant variation in WVP for TPS film when treated with HPP	TS of the thermally processed TPS film 24.67 ± 1.03 MPa significantly increased to 26.92 ± 0.43 MPa upon application of 600 MPa	EAB of the thermally processed TPS film 5.04 ± 0.56% significantly increased to 5.71 ± 0.20% when subjected to 600 MPa	T_o_, and ΔH of thermally processed TPS increased from 70.92 °C and 56.92 J/g to 84.32 °C and 78.40 J/g, respectively but T_m_ decreased from 124.62 to 122.07 °C; upon application of 600 Mpa	
PVA, chitosan (CHI), and nano-TiO_2_	200, 400, and 600 MPa at 23 ± 2 °C for 15 min	--	WVP of PVA–CHI–TiO_2_ (0.10%) (4.36 ± 0.308) × 10^−12^ g·cm/cm^2^·s·Pa significantly decreased to (3.60 ± 0.137) × 10^−12^, (3.47 ± 0.139) × 10^−12^, and (3.92 ± 0.0433) × 10^−12^ g·cm/cm^2^·s·Pa when subjected to 200, 400, and 600 MPa, respectively; OP of the film 1.34 ± 0.05 cm^3^ m^−2^·s^−1^·Pa^−1^ showed no significant variation when treated with 200 MPa but OP significantly decreased to 1.30 ± 0.05 and 1.25 ± 0.05 cm^3^ m^−2^·s^−1^·Pa^−1^ when treated with 400 and 600 MPa	TS of PVA–CHI–TiO_2_ (0.10%) 8.24 ± 0.27 MPa significantly increased to 13.67 ± 0.41, 13.98 ± 0.33, and 17.15 ± 0.97 when subjected to 200, 400, and 600 MPa, respectively	EAB of PVA–CHI–TiO_2_ (0.10%) 64.82 ± 1.10% significantly increased to 68.48 ± 1.66, 68.12 ± 1.94, and 67.92 ± 2.73% when subjected to 200, 400, and 600 MPa, respectively	--	[36]
Chitosan	100, 200, 300, 400, and 500 MPa for 15 min	--	WVP and OP of the chitosan film decreased continuously when the pressure increased from 100 to 500 MPa	TS of film increased 35.2% as compared to the untreated film when treated at 400 MPa for 15 min but further increase in the pressure decreased the TS	EAB of the chitosan film decreased continuously as the pressure increased from 100 to 500 MPa	--	[45]
Pigskin gelatin	0.1, 300, and 600 MPa at 20, 40, and 60 °C for 5, 17.5, and 30 min	--	WVTR of the untreated film 65.56 ± 1.2 g/(day m^2^) significantly decreased to 63.47 ± 0.9 g/(day m^2^), when subjected to 600 MPa for 30 min at 20.5 °C	TS of the untreated film 25.7 ± 2.2 MPa significantly increased to 28.7 ± 2.5 MPa when subjected to 600 MPa for 30 min at 20.5 °C	EAB of the untreated film 8.6 ± 0.6% insignificantly increased to 10.1 ± 1.5% when subjected to 600 MPa for 30 min at 20.5 °C	T_g_, and T_m_ of the untreated film 58.8 ± 0.4, and 131.5 ± 0.7 °C increased to 60.7 ± 4.5, and 138.2 ± 0.5 °C, respectively, but ∆H_m_ decreased from 46.4 ± 0.8 to 36.5 ± 3.3 J/g, subjected to 600 MPa for 30 min at 20.5 °C	[41]
Amaranth protein	200, 400, and 600 for 5 min	WS of the untreated film 79.9 ± 2.1% significantly decreased to 56.4 ± 5.5, 46.1 ± 0.5, and 46.1 ± 2.5% when treated with 200, 400, and 600 for 5 min, respectively	WVP of the untreated film (5.6 ± 0.5) × 10^−12^ g H_2_O/Pa m s significantly decreased to (4.8 ± 0.4) × 10^−12^, (4.6 ± 0.1) × 10^−12^, and (3.2 ± 0.6) × 10^−12^ g H_2_O/Pa m, when treated with 200, 400, and 600 for 5 min, respectively	TS of the control film increased by 26%, 101%, and 165% when subjected to 200, 400, and 600 for 5 min, respectively	No significant variation in EAB under high-pressure treatment	--	[37]
Nisin-soy-protein-isolate	100, 200, 300, 400, and 500 MPa at 20 °C for 10 min	--	WVP of the untreated film significantly decreased as the pressure level increased from 100 to 500 MPa	TS of the untreated film significantly increased as the pressure level increased from 100 to 500 MPa	EAB of the untreated film significantly decreased as the pressure level increased from 100 to 500 MPa	--	[30]
Whey protein concentrate, thyme (TEO)	600 MPa at 70 °C, for 20 min	--	WVP of thermally treated WPC-TEO film was (24.867 ± 2.855) × 10^−10^ g/s.m.Pa significantly decreased to (10.178 ± 1.690) × 10−10 g/s.m.Pa, when subjected to 600 MPa at 70 °C, for 20 min	--	--	--	[50]
Poly (lactic acid) and Ag (5%)	0, 200, and 400 MPa for 15 min at 25 °C	--	WVP of untreated PLA/Ag-5% film (4.3 ± 0.3) × 10^−10^ (g·m/m^2^·s·Pa) significantly decrease to (2.8 ± 0.1) × 10^−10^ and (3.2 ± 0.2) × 10^−10^ (g·m/m^2^·s·Pa), when subjected to 200 and 400 MPa for 15 min	TS of untreated PLA/Ag-5% film 34 ± 2 MPa significantly increased to 36 ± 2 MPa at 400 MPa for 15 min	EAB of untreated PLA/Ag-5% film 170 ± 8% significantly decreased to 161 ± 14 and 119 ± 14%, when subjected to 400 MPa for 15 min	T_g_, and T_c_ of PLA/Ag-5% film 50.1 ± 0.2, and 110.4 ± 0.4 °C significantly decreased to 51.9 ± 0.2, and 112.9 ± 0.5 °C, respectively, when treated with 400 MPa for 15 min; T_m_ showed no significant variation between treated and untreated film	[32]
Poly (lactic acid) and ZnO (0, 2.5, 5.0 and 10.0 % of PLA)	0, 200 and 400 MPa for 10 min	--	OP of the untreated PLA/ZnO-5% film 4.83 ± 0.13 (cm^3^ 24 h^−1^ m^−2^) × (cm bar^−1^) slightly decreased to 3.02 ± 0.29 (cm^3^ 24 h^−1^ m^−2^) × (cmbar^−1^) when subjected to 400 MPa for 10 min.; WVP of the PLA/ZnO-5% film decreased significantly when subjected to 400 MPa for 10 min.	TS of untreated PLA/ZnO-5% film 35.8 ± 1.48 MPa, increased to 41.9 ± 1.43, and 42.9 ± 1.08 MPa when subjected to 200, and 400 MPa for 10 min, respectively	EAB of untreated PLA/ZnO-5% film 8.19 ± 0.17% decreased to 7.90 ± 0.34, and 7.61 ± 0.58% when treated with 200, and 400 MPa for 10 min, respectively	T_g_ and T_c_ of untreated PLA/ZnO-5% film 46.7 ± 1.82 and 95.9 ± 0.30 °C significantly increased to 49.8 ± 1.50 and 100.9 ± 0.70 °C and showed no significant variation in T_c_ when subjected to 400 MPa for 10 min	[42]

TS: Tensile strength; EAB: Elongation at break; R: Reference.

## 3. Effect of HPP on the Properties of Flexible Biopolymer-Based Packaging Materials

While the previous section discussed the physical properties of the film obtained from HPP of film forming solution (FFS). This section elaborates on the impact of HPP on the properties of the preformed flexible biopolymer-based packaging materials. Currently, HPP has been extensively used in food processing and preservation of aquatic food, meat, dairy, fruits, and vegetables. It is mostly focused on the sterilization effect of food microbes and the quality change of food products [33,51]. For the packaging materials to be used in the HPP environment, several factors like pressure resistance, water, and oxygen barrier property, restriction to the leaching, seal integrity, and clarity have to be considered. When exposed to HPP, volume compression (approximately 19%) and equal expansion upon decompression occur and thus packaging materials should be able to withstand the tremendous pressure variation retaining the seal integrity, mechanical strength, and sufficient barrier properties [51]. Generally, flexible packaging materials experience reversible changes during HPP [4,52]. However, the irreversible changes cause visible deformation and variations in the functional properties of the packaging materials. A probable elucidation for irreversible deformation could be due to the fact that during HPP, gases are adsorbed within the layers of the film matrix due to compression and instantly released in the course of depressurization. During compression, gas may act as a plasticizer and dissolve easily in the inner film resulting in the change in crystallinity of the polymers. Also, an instantaneous increase in specific volume during the rapid depressurization may cause the gas bubbles to expand fast and burst leading to the development of cavities [4,51,53]. This compression and decompression leads to the deviation in the properties of the HPP-treated packaging materials.

### 3.1. Surface Attributes (Morphological Characteristics)

Food packages when subjected to HPP should be able to retain properties similar to those manufactured, evading defects such as delamination, formation of wrinkles, holes, or other defects, and should maintain their visual integrity. Such defects, and the deviation of built-in properties of the materials, may result in changes in aesthetic qualities including their design or dimension, compromising the safety and shelf life of food products that can even result in the rejection of the product [4]. Besides visual defects that are detected with naked eyes, the biopolymer-based film may also experience microscopic defects and should be assessed. The most widely used techniques for microscopic analysis include SEM or atomic force microscopy (AFM).

The visual defects, such as bubbles, embrittlement, and opacity are observed in the HPP of biopolymer-based packaging materials. Sansone, Aldi [54] investigated the suitability of the commercial PLA-based flexible packaging (*Biophan 121*) for HPP (pasteurization and sterilization of carrot-based products). The pasteurization (25–40 °C), and sterilization (90–115 °C) were carried out at different pressure levels of 200, 500, and 700 MPa for 5 min. HP pasteurization showed no significant variation in the structural or functional properties of the *Biophan 121.* However, visual inspection showed that HP sterilization at 700 MPa at 115 °C for 5 min caused unacceptable embrittlement and opacification of *Biophan 121,* making it unsuitable for HP sterilization applications as shown in Figure 4. 

Galotto, Ulloa [12] studied the effect of HPP (500 MPa at 50 °C for 15 min) on the physical properties of PLASiOx/PLA films and reported superficial modifications in the structure of the films filled with water (aqueous simulant), and olive oil (fatty food simulant). SEM images (Figure 5) showed that HPP affected the integrity of the inorganic coating of the PLASiOx/PLA film. Some pinholes were observed in the film when in contact with both aqueous and fatty food simulant but major damage was noticed in the case of aqueous simulant in the form of bright areas which could be due to swelling as shown in Figure 5B. Such swelling may result in undesirable changes like a large reduction in the barrier properties [12]. 

Figure 6 shows the cross-sectional SEM micrographs (×1000 magnification) of PLA-PEG and PLA-PEG-GO-2% film subjected to HPP. The cross-sectional micrographs showed HPP at 450 MPa developed cracks on the surface and exhibited roughness. While the film developed small holes and non-uniform cavities when subjected to 600 MPa. This could be due to the entrapment of air that acts as a plasticizer and is dissolved in the inner layer of the film during compression leading to the development of roughness in the film. Also, at rapid depressurization, the entrapped gas bubbles might have experienced fast expansion and bursting in the inner matrix of the film resulting in the creation of holes and cavities [51]. Such observations in the microstructure of the PLA-based film are reported for HPP of PLA-Ag nanocomposite films [33], PLA-TiO_2_ nanocomposite film [55], and PLA-Ag nanocomposite film [32]. Other biopolymer-based packaging materials such as cellulose acetate films when subjected to HPP undergo significant changes in the structure such as swelling, delamination, and formation of holes and cracks after processing [50,56]. 

Conversely, Ahmed, Mulla [57] reported that HPP (450, and 600 MPa) had no adverse effect on the co-extruded PLA film having a thickness of 25 and 75 μm. The topographic surface obtained from AFM of the film showed that surface roughness parameters (arithmetic mean height: *S_a_* and root means square height: *S_q_*) of the 25 μm co-extruded PLA film significantly decreased indicating the formation of a smoother surface upon HPP and showed no significant variation in the case of thick film 75 μm. Such a variation in the surface roughness of two coextruded PLA samples indicates that the thickness or more specifically the architecture of the film is a significant factor when subjected to HPP [57]. 

### 3.2. Barrier Properties

In food packaging, barrier properties (importantly water vapor and oxygen barrier properties) are key parameters in the selection of packaging materials as they play a significant role in maintaining the quality attributes of foods. Packaging materials subjected to HPP should be able to withstand the changes caused due to compression and instant depressurization during processing. As per industrial norms, a deviation up to 12% in barrier property is acceptable without compromising their integrity and performance [1]. In general, biopolymer-based packaging materials have lower barrier properties as compared to conventional packaging materials [47]. Therefore, the selection of biopolymer-based packaging material should be in such a way that when exposed to HPP, barrier properties are enhanced or at least do not change. Thus, some studies have examined the effect of different conditions of HPP on the barrier properties of biopolymer-based films. 

Currently, PLA is the most frequently used commercial biopolymer-based packaging material for HPP. Some of the studies have reported that HPP increases the WVP and OP. The extent of modification of barrier properties of biopolymer-based film subjected to HPP depend on the pressure level applied. It can be observed that with the enhancement in the pressure levels from 0 to 600 MPa, the WVP of PLA-PEG-GO-2% nanocomposite film increased from (1.05 ± 0.11) × 10^−14^ (kg m) (m² s Pa) to (1.68 ± 0.21 × 10^−14^) (kg m) (m² s Pa). This increase in the WVP can be attributed to the plasticizing effect of water on polyethylene glycol in an aqueous HPP environment. This enhancement in WVP could also be due to the formation of holes and cavities on the film structure during HPP. Such an increasing trend in WVP with the increase in pressure level was observed in the case of co-extruded PLA films, where, the WVP of the untreated film (1.55 ± 0.12) × 10^−14^ (kg·m/[m^2^·s·Pa]) increased to (1.59 ± 0.10) × 10^−14^ and (1.62 ± 0.13) × 10^−14^ (kg·m/[m^2^·s·Pa]) when subjected to 450 and 600 MPa [58]. The authors state that the reduction in the crystallization of the co-extruded film might have facilitated the transport of water vapor and thus enhanced the permeability. A similar increase in WVP was also noticed in the case of PLA-Ag NPs-based nanocomposite films when exposed to 400 MPa for 20 min [33]. 

Similar to the WVP, the application of different pressure levels had a significant impact on the OP of the biopolymer-based film as summarized in Table 2. It was observed that with the increase in the pressure level from 0 to 600 MPa, OP of PLA-PEG-GO-2% nanocomposite film increased from 2.18 ± 0.12 × 10^−18^ to 6.54 ± 0.37 × 10^−18^ (kg m) (m² s Pa). This increase in OP was attributed to the decrease in the crystallinity PLA-PEG-GO-2% nanocomposite film under HP treatment [51]. Such an increasing trend in the OP with the increase in pressure level was also reported for co-extruded PLA film, where the OP of the untreated 25-μm film significantly increased from (6.55 ± 0.41) × 10^−18^ to (9.67 ± 0.84) × 10^−18^ [kg·m/ (m²·s·Pa)] [57]. Similarly, Ahmed, Mulla [59] also reported an increase in oxygen transmission rate (OTR) in PLA-based film loaded with cinnamon essential oil (CIN). The application of 300 MPa, at 23 °C for 10 min on PLA-CIN might have resulted in the structural alteration, facilitating the oxygen transmission. 

Besides pressure level, processing temperature also plays a vital role in the alteration of barrier properties of biopolymer-based films. It was observed that HP sterilization (700 MPa, at 90–110 °C) of carrots using flexible PLA film was not suitable as the film was damaged during HP sterilization, as shown in Figure 4. However, HP pasteurization of carrot using PLA flexible film demonstrated an enhancement in water vapor barrier properties and this could be due to the structural alteration of the film matrix during HPP processing at 25 and 30 °C [54].

Further, it was observed that the impact of HPP on the barrier properties of the biopolymer-based film is also dependent on the types of food materials being packed. As observed in PLASiOx-PLA film, application of 500 MPa at 50 °C for 15 min, OP increased by 31% when in contact with olive oil (fatty acid simulant) but OP was so large in case of the film contact with distilled water (aqueous simulant) that the film could not be used for food protection. Similar to the OP, WVP was also affected by the types of food being packed, for instance, WVP increased by 2170% when the film was in contact with distilled water and 71% when in contact with the olive oil compared to the control film. This could be due to the plasticizing effect of water on hydrophilic PLA leading to the swelling along with the creation of holes and cavities when subjected to high pressure as explained in Section 3.1 [12]. Such an increase in WVP and OP was observed in the case of co-extruded PLA film when in contact with water with an increase in pressure level from 0 to 600 MPa.

Some other studies reported that HPP has no significant impact on the barrier properties of the film, for instance, 75-μm coextruded PLA film subjected to HPP (300, 450, and 600 MPa) [57], PLA/Nano-TiO_2_ nanocomposite treated with 300 MPa [60]. However, a few studies have also demonstrated that HPP enhances barrier properties. For instance, the WVTR of cellulose acetate film decreased with the increase in pressure level (200, 300, and 400) and exposure time (5 and 10 min) [56] as summarized in Table 2. Similarly, WVP and OP of PLA-TiO_2_ nanocomposite film significantly decreased when subjected to HPP (300 MPa) [55]. Such an increase in the barrier properties was attributed to the increase in the crystallinity of the composite film due to HPP, where the film matrix is more closely arranged and thus prevents the transportation of oxygen and water vapor.

Overall, PLA film prepared by solvent casting method or by extrusion method and subjected to HPP increases the barrier property due to reduction in crystallinity of the film. Finally, it can be concluded that biopolymer-based films required serious modification for further enhancement for commercial applications.

### 3.3. Mechanical Properties

Among the biopolymers, PLA-based packaging is mostly used for the HPP of food materials at the commercial level. Only a few studies have shown the application of other biopolymers such as cellulose acetate and wheat gluten-based packaging systems for HPP. The mechanical properties of the biopolymer-based packaging system depend on the type of polymer matrix, different additives, and their interactions. It has been observed that HPP can lead to the modification of biopolymer-based packaging materials and is dependent on the processing conditions such as pressure level, temperature, and time of exposure. Only a few studies were found in the literature that reported the enhancement in mechanical properties of biopolymer-based packaging systems when subjected to HPP. It either reduces the mechanical properties of the film or in some cases, the materials remain unaffected as summarized in Table 2. 

The extent of modification of mechanical properties of the biopolymer-based film subjected to HPP is dependent on the level of pressure. It was observed that TS of untreated PLA-PEG-GO-2% film 50.80 ± 0.75 MPa decreased to 43.13 ± 6.64, 40.69 ± 0.77, and 40.14 ± 1.00 MPa when subjected to 300, 450, and 600 MPa, respectively. Similarly, EAB of the untreated nanocomposite film 25.31 ± 0.27% decreased to 20.32 ± 1.35, 17.98 ± 0.92, and 11.72 ± 1.35% when subjected to 300, 450, and 600 MPa, respectively [51]. Such a decrease in TS and EAB might be attributed to the formation of cracks and cavities in the film structure during HPP [51]. Besides pressure level, the time of processing is also an important factor that can influence the mechanical properties of flexible packaging materials. The application of 200, 300, and 400 MPa for different periods such as 5 and 10 min had a different impact on the mechanical properties of the cellulose acetate film. For instance, TS of untreated cellulose acetate film was 40.9 ± 1.2 MPa when subjected to 300 MPa for 10 min, TS was 36.6 ± 1.3 MPa but the reduction in TS was higher (28.9 ± 1.3 MPa) when subjected to HPP for 5 min. This reduction could be due to changes in crystallinity, film delamination, or plastification caused by HHP [56]. It was also observed that the EAB of HP-treated PLA-Ag nanocomposite film decreased with the increase in migration time, although TS was not altered [33]. In contrast, Chi, Li [55] reported that TS increased with the increase in Ag NPs concentrations (from 0 to 20 wt%) when subjected to 300 MPa but EAB decreased. The increase in TS could be due to the increased stiffness of the composite film during HPP [55]. 

Literature also reports that the extent of modification of flexible packaging materials subjected to HPP also depends on the types of food being packed. The application of HP into the film containing liquid food can result in plasticization of the material that weekends the structure leading to a reduction of mechanical properties. Notably, plasticization of polymers occurs in the amorphous region rather than the crystalline region. Such effect was evident in the low crystalline PLASiOx-PLA film, where the film in contact with olive oil (fatty acid simulant) experienced higher degradation in mechanical properties as compared to film contact with distilled water (aqueous simulant). PLASiOx-PLA film when subjected to 500 MPa at 50 °C for 15 min decreased TS and EAB by almost 25% and 32%, respectively when in contact with fatty acid simulant and 22% and 29%, respectively when in contact with aqueous simulant [12]. 

Some studies have reported that HPP had a minor to no significant impact on the mechanical properties of the biopolymer-based films. Ahmed, Mulla [59] investigated the synergistic effect of HPP and PLA-PEG-based active film on refrigerated storage of the chicken sample. It was reported that the application of 300 MPa at 23 °C for 10 min had no significant impact on the TS and EAB of PLA-PEG film and PLA-PEG-CIN-17% film. Similarly, Fan, Cui [33] studied the effect of Ag nanoparticle content (1, 5, 10, 15, and 20 wt%), HPP (100/200/300/400 MPa for 10/20/30 min), and the migration time (0, 14, 28, 49 days) and reported that HPP had no significant impact on the TS, although EAB was decreased with the increase in migration time. 

### 3.4. Thermal Properties

As mentioned in Section 2.5, the DSC method is one of the most widely used techniques to determine the thermal properties of packaging materials. Table 2 summarizes the thermal properties of HPP of biopolymer-based films. The HPP has different effects on the thermal characteristics of biopolymer-based films and in some cases, thermal stability is increased. Ahmed, Mulla [51] investigated the impact of HPP on the thermal behavior of PLA-PEG-GO nanocomposite film. The results demonstrated that HPP (0, 300, 50, and 600 MPa) prompted the T_g_, T_m_, T_c_, ΔH_m_, and ΔH_c_ values of PLA-PEG-GO-1% GO nanocomposite film, showing the postponed thermal degradation compared to the untreated film. Such enhancement in the thermal stability could be attributed to the increase in the crystallinity of the nanocomposite due to HPP. Such enhancement in thermal stability was also observed for PLA film subjected to HP sterilization (700 MPa at 90 °C), due to crystallization of the amorphous external layer PLA matrix. Crystalline regions can act as mobility constraints for the macromolecules in the amorphous phase, thus enhancing thermal stability [54]. 

Besides pressure and temperature, different foods/food simulants in contact with film subjected to HPP, significantly influence the thermal properties. As evident from PLASiOx-PLA film when subjected to HPP (500 MPa for 15 min at 50 °C), the T_m_ value of the film in contact with oil simulant showed no significant difference but decreased when contact with water. Such variation was ascribed to the redistribution in the crystallite size when in contact with a liquid of a great affinity to PLA. Where the percentage crystallinity of the untreated film was 2.4% decreased to 1.7 and 1.9%, respectively when in contact with oil and aqueous simulant, respectively [12]. Some other studies have also reported that with the application of HP, thermal stability decreased. For instance, in the case of low crystallinity, polymeric materials like cellulose acetate film T_g_ and T_m_ significantly decreased when exposed to 200 and 300 MPa for 5 and 10 min [56]. In another study, Tang, Fan [60] showed that T_g_ and T_c_ values of PLA-TiO_2_ nanocomposite film increased when subjected to HPP, but showed no significant variation in T_m_ value after HPP. 

However, some studies have demonstrated that HPP had a minor no significant variation in the thermal properties of the film. For instance, PLA-PEG-cinnamon oil-4% film subjected to 300 MPa showed no significant variation in thermal properties, indicating PLA-PEG-cinnamon oil-4% was pressure resistant [59]. Similarly, Chi, Li [55] also reported that T_g_, and T_c_, of the PLA-TiO_2_ nanocomposite film subjected to 300 MPa for 10 min showed no significant variation, however, % crystallinity increased significantly for the HP treated film. Similarly, no significant variation in T_m_ was observed for co-extruded PLA film subjected to HPP (600 MPa) [57]. 

### 3.5. Migration Potential

Migration can be defined as the diffusion of substances from a region of greater concentration (the food-contact-surface) to one of a lower concentration (usually the food surface). This process is often influenced by food-packaging interactions. The primary packaging material that comes in direct contact with food material interacts through mass transfer mechanisms like permeation, sorption, and migration [4,64,65]. Among other substances, low molecular weight compounds, organic solvents, plasticizers, antioxidants, and monomers are the main substance that migrates from the packaging to the food and can affect the sensory attributes and level of toxicity of the packaged food materials. Such migration must be controlled as these substances can be toxic and harmful to humans, animals, and the environment. Therefore, the primary packaging materials must be assessed to confirm that the packaging system poses no threat to the food contained within it and avoid being the source of contamination, whether of physical, chemical, or biological nature [4,65,66]. Among others, chemical contamination from packaging materials can be due to the presence of different components in the polymer matrix and when subjected to HPP, the migration potential must be studied. In Europe, biopolymer-based packaging materials that come in contact with food are regulated following Regulation (UE) No. 10/2011 used for conventional plastic packaging [67]. As per this regulation, any potential risk in the final product associated with the migration of any harmful compounds should be assessed by the manufacturer according to the internationally recognized scientific principles on risk assessment. 

Some of the studies have shown that HPP of the biopolymer-based film reduced the migration of packaging additives into food products as well as reduced the absorption capacity of the packaging materials. For instance, Mauricio-Iglesias, Jansana [61] investigated the migration of a packaging additive Uvitex OB^®^ (Florham Park, NJ, USA)from a PLA matrix and found that migration was so low that no variation in the initial concentration of the additives was detected in PLA film subjected to HPP (800 MPa at 20 to 40 °C for 5). The authors reported that approximately 0.03% *w*/*w* of Uvitex OB^®^ was detected for the HP-treated PLA film in contact with four food stimulating liquids distilled water, 3% acetic acid, 15% ethanol, and olive oil, which is very low from a specific migration limit of 0.6 mg/kg. Interestingly, Mauricio-Iglesias, Peyron [62] reported that HPP (800 MPa for 5 min) at 40 °C favored the decrease of free volume in PLA films, ensuing in a substantial reduction of aroma adsorption by almost 50% as compared to the untreated film. However, HPP (800 MPa for 5 min) at 115 °C enhanced the aroma adsorption.

However, some studies have also demonstrated that HPP may induce the migration of packaging additives. For example, Tang, Fan [60] investigated the influence of HPP (300 MPa for 10 min) on the total migration of TiO_2_ in a 50% (*v*/*v*) ethanol solution as a food simulant. As shown in Figure 7, the total migration of TiO_2_ increased with the increase in the concentration of nanoparticles (1 to 20 wt.%) and migration time (0 to 45 days). The highest concentration of TiO_2_ in the ethanol solution was found to be 0.43 ± 0.01 mg/kg for HP-treated nanocomposite film soaked for 45 days. Interestingly, Yang, Zhu [68] reported that the highest migration content of TiO_2_ NPs from untreated PLA-TiO_2_ nanocomposite after 45 days of soaking in 50% (*v*/*v*) ethanol solution was 0.54 ± 0.04 mg/kg. The lower rate of migration of TiO_2_ NPs from HP-treated film could be due to the increase in the crystallinity of the composite film under the influence of HPP.

Similarly, Fan, Cui [33] examined the migration of Ag NPs from PLA-Ag nanocomposite film matrix treated with different pressure levels and treatment times. Migration of Ag NPs increased with the increase in the nanoparticles concentration (from 10 to 20%), treatment time (10 to 30 min), and migration time (0 to 49 days). The largest migration concentration of Ag NPs was found to be 29.5 mg/kg when PLA-Ag-20% nanocomposite film at 40 °C was treated with 200 MPa for 30 min and soaked in isooctane as a food simulant for 49 days. Such a performance posed a threat to the application of biopolymer-based films loaded with packaging additives, and thus investigating film property changes under HPP is necessary.

## 4. Future Perspective

From the literature review, it has been observed that the application of HPP to the film-forming solution (FFS) can form much denser film and thus enhance the properties of the developed packaging materials. However, only a few studies have been carried out to investigate the impact of HPP on biopolymer-based films and most of the research is focused on the effect of HPP on the development of film from a single polymer. As stated, HPP may induce changes in the non-covalent bonding between the components of films resulting in the performance of the packaging materials. Therefore, composite packaging material such as a combination of polysaccharide-polysaccharide, protein-protein, polysaccharide-protein, and/or their combination with some functional compounds needs to be further examined. 

The petroleum-based packaging has been used for HPP of food products and its impact on the packaging properties is well established, while limited information is available on biopolymer-based packaging. Moreover, the results reported in the scientific literature show inconsistent results, and thus the understanding of the impact of HPP on biopolymer-based packaging material is rather limited. Therefore, further research is required to understand the post-performance of HP-treated biopolymer-based films. More research on packaging properties such as mechanical, thermal, barrier, and the integrity of the post-treated films should be performed in the lab and pilot-scale to optimize biopolymer-based packaging materials. Moreover, many additives such as nanoparticles or functional components are added to enhance the packaging properties and the application of HP may induce changes in the interaction between these components, and thus different migration rates into food may pose threat to human safety. Therefore, the migration of these novel materials from packaging material into food product needs to be evaluated and develop reliable models to predict its migration concentration during the course of processing, storage, and distribution. Furthermore, HPP and active packaging are suitable options for food packaging applications and the release kinetics of active compounds from biopolymer-based packaging materials under HPP is also an area to be explored.

## 5. Conclusions

In this review, the effect of HPP on the functional properties of the biopolymer-based film was studied including the application of HP to the film-forming solution (FFS) prior to drying and HPP at low and high temperatures in foods or food simulants packaged in biopolymer-based film. The literature revealed that HPP of FFS, formed film with superior properties such as compact and uniform film, enhanced mechanical, thermal, and barrier properties compared to the untreated films. However, the application of HP treatment to the performed biopolymer-based packaging materials experiences irreversible structural damage causing a reduction in the functional properties of the packaging materials. HPP of biopolymer-based film at a relatively lower temperature (pasteurization) induces deviation in the packaging properties in the acceptable range. However, when exposed to HPP at high temperatures such as sterilization temperature, opacification, and embrittlement are observed in the film, and thus restrict their commercial applications.

Currently, only a few studies have been carried out focusing on the effect of HPP on the structural, functional, and migration potential of biopolymer-based films. Moreover, the outcome of these studies is not homogeneous, which demands more technical advancement to achieve a deeper understanding of the real effects that HPP can cause on biopolymer-based films. Thus, industries can avail greater support in developing biopolymer-based packaging films having the desired properties for the HPP of foods and beverages.

The information regarding HPP of synthetic plastics has already been consolidated to some extent, such as the desired properties of packaging materials for HPP of food products like flexibility, strength, dimensional stability, and heat seal integrity. This information can be taken into consideration in developing biopolymer-based film with the necessary requirements for HPP and a thorough study is required to evaluate their deviation during processing, storage, distribution, and consumption of processed food. Overall, the available limited information regarding biopolymer-based film demands complete studies to develop a biopolymer-based environmentally friendly packaging alternative for HPP.

## Figures and Tables

**Figure 1 polymers-14-03009-f001:**
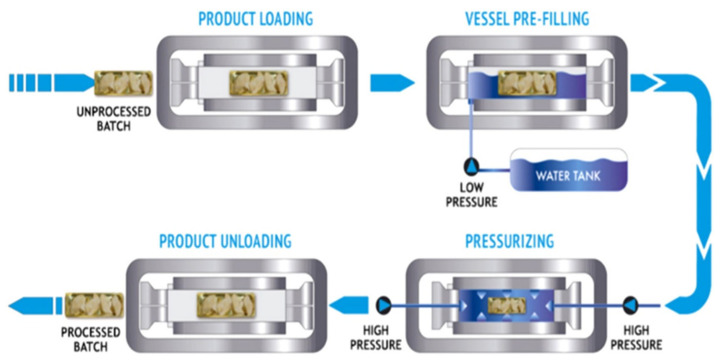
Diagram of high-pressure processing (HPP), Reprinted with permission from Ref. [4], Copyright (2022) Elsevier.

**Figure 2 polymers-14-03009-f002:**
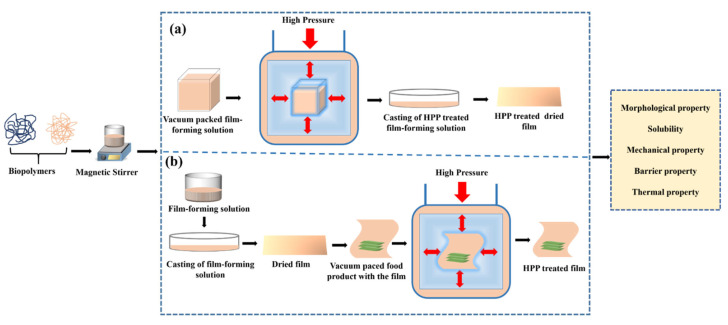
A graphical representation of the application of HPP on the films: (**a**) HPP applied to film-forming solution; (**b**) HPP applied on the dried film.

**Figure 3 polymers-14-03009-f003:**
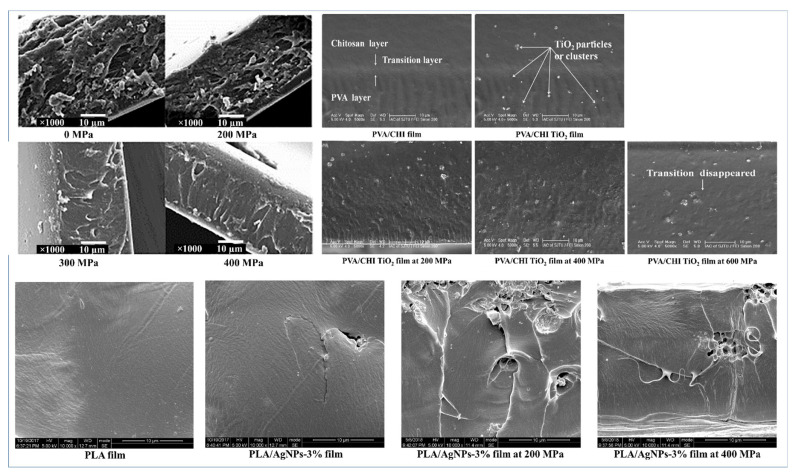
Micrographs of HPP treated biopolymer-based film when applied before casting of films: Cross section of HPP (0, 200, 300, and 400 MPa) treated nisin-soy-protein-isolate film, reprinted with permission from Ref. [30], copyright (2022) Elsevier; Cross section of HPP (0, 200, 400, and 600) treated PVA/CHI/TiO2 film, reprinted with permission from Ref. [36], copyright (2022) Elsevier; Cross section of HPP (at 0, 200 and 400 MPa) treated PLA/AgNPs-3% film, reprinted from Ref. [32].

**Figure 4 polymers-14-03009-f004:**
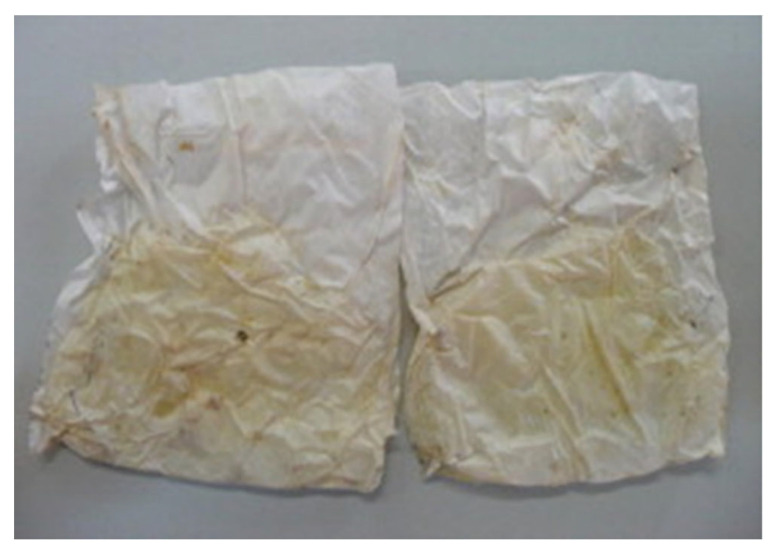
Pictograph of a PLA *Biophan 121* pouches after HP sterilization (700 MPa at 115 °C for 5 min) containing solid carrots as packaged foodstuff, reprinted with permission from Ref. [54], copyright (2022) Elsevier.

**Figure 5 polymers-14-03009-f005:**
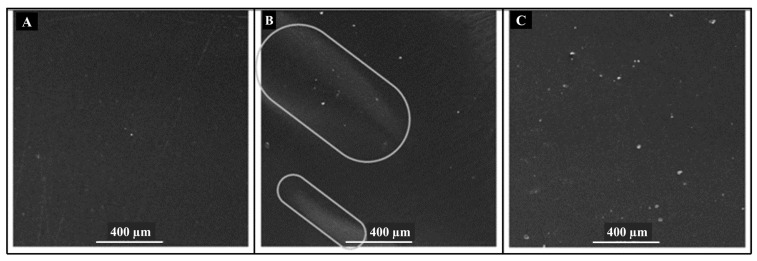
SEM micrographs PLASiOx-PLA films: (**A**) Untreated film, (**B**) HPP treatment at 500 MPa and 50 °C for 15 min, in contact with water as food simulant, and (**C**) HPP treatment at 500 MPa and 50 °C for 15 min, in contact with olive oil as food simulant, Reprinted with permission from Ref. [12], Copyright (2022) Elsevier.

**Figure 6 polymers-14-03009-f006:**
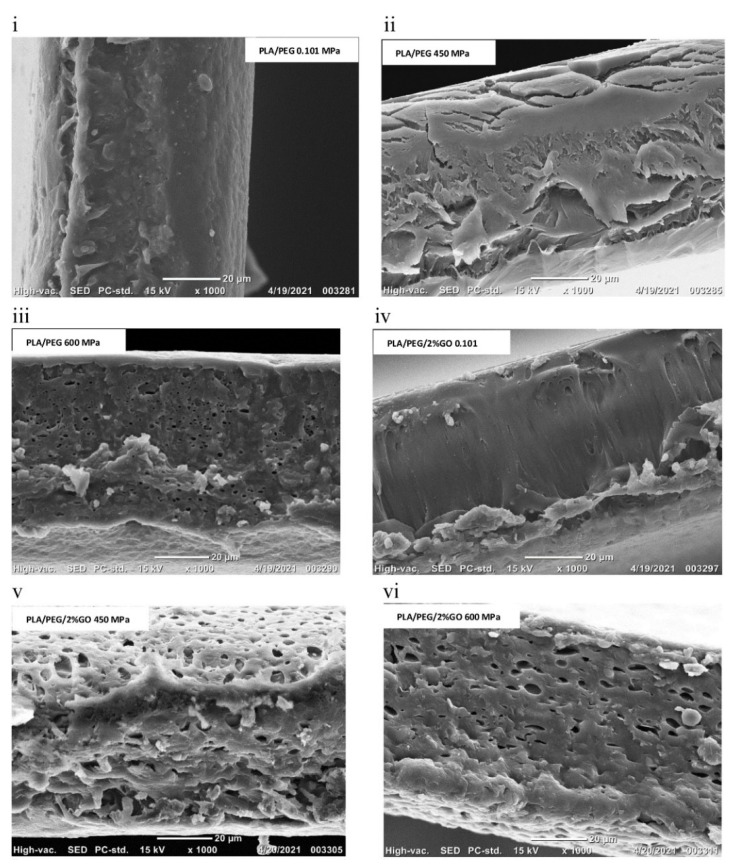
SEM cross sections of the PLA-based films PLA/PEG and PLA/PEG/2% GO films: (**i**) PLA/PEG at 0.101 MPa; (**ii**) PLA/PEG at 450 MPa; (**iii**) PLA/PEG at 600 MPa; (**iv**) PLA/PEG-2% GO at 0.101 MPa; (**v**) PLA/PEG-2% GO at 450 MPa; (**vi**) PLA/PEG-2% GO at 600 MPa (×1000 magnification), Reprinted with permission from Ref. [51], Copyright (2022) Elsevier.

**Figure 7 polymers-14-03009-f007:**
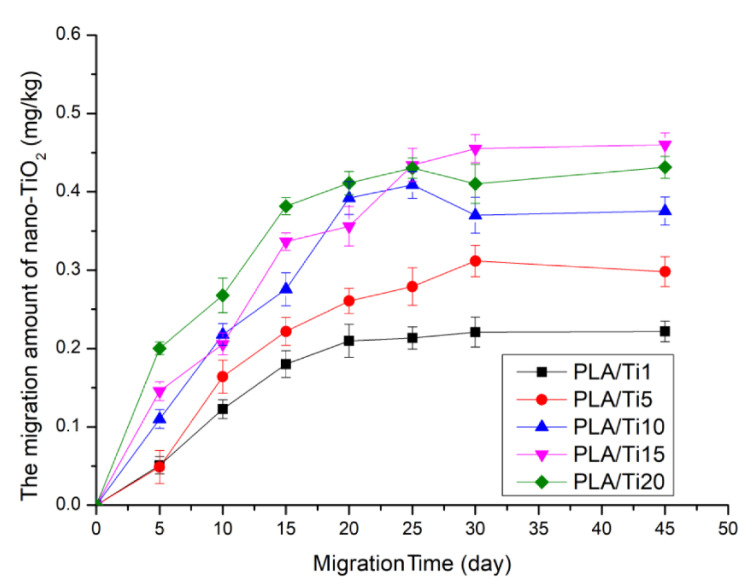
The concentration of TiO_2_ NPs migrated from PLA-TiO_2_ nanocomposite film to 50% (*v*/*v*) ethanol solution as food simulant after high-pressure treatment of 300 MPa for 10 min in a migration period of 0 to 45 days, Reprinted from Ref. [60].

**Table 2 polymers-14-03009-t002:** Effect of HPP on the physical properties of the biopolymer-based films.

Film Matrix	Processing Conditions	Food Simulant	Barrier Property (WVP/OP)	Mechanical Property		Thermal Properties	Additive Migration	R
				Tensile Strength (TS)	Elongation at Break (EAB)			
PLA, PEG, and CIN	200, 250, and 300 MPa at 23 °C for 10 min	Chicken	OTR of untreated PLA-PEG-CIN 4% film was 725.63 ± 20.00 (ml/m^2^ day) significantly increased to 771.58 ± 18.50 (ml/m^2^ day) when subjected to 300 MPa/23 °C /10 min	TS of untreated PLA-PEG-CIN 4% film was 10.08 ± 0.14 MPa and showed no significant variation when subjected to 300 MPa/23 °C /10 min and TS value was 9.82 ± 0.90 MPa	EAB of untreated PLA-PEG-CIN 4% film was 100.55 ± 4.51% and showed no significant variation when subjected to 300 MPa/23 °C /10 min and the EAB value was 104.64 ± 5.59%	T_g_ of untreated PLA-PEG-CIN 4% film was 1.44 ± 0.01 °C and showed no significant variation when subjected to 3300 MPa/23 °C /10 min; T_m_ significantly increased from 136.97 ± 0.14 to 137.59 ± 0.42 °C but H_m_ showed no significant variation; Tc and % χ_c_ significantly decreased from 62.20 ± 0.91 to 60.79 ± 0.78 °C and 11.03 ± 0.39 to 9.86 ± 0.25%, respectively when subjected to 300 MPa/23 °C /10 min	--	[59]
PLASiOx/PLA	500 MPa at 50 °C for 15 min	Olive oil, Distilled water	WVP of the untreated film increased by 2170.0% when in contact with water and 71.0% when in contact with olive oil. OTR of the untreated film enhanced by 31.0% when in contact with olive oil and the increment was too large when in contact with distilled water making the film unsuitable for packaging	TS of untreated PLASiOx-PLA was 101.1 ± 4.6 MPa significantly decreased to 75.4 ± 2.5 MPa for HPP–oil simulant and 79.2 ± 4.4 MPa for HPP–aqueous simulant	EAB of untreated PLASiOx-PLA was 4.1 ± 0.2% significantly decreased to 2.8 ± 0.1% for HPP–oil simulant and 2.9 ± 0.2% for HPP–aqueous simulant	T_m_ of untreated PLASiOx-PLA was 148.3 °C significantly increased to 148.6 °C for HPP-oil simulant and but decreased to 146.0 °C for HPP in contact with aqueous simulant; ΔH_m_ of the untreated film decreased from 11.9 to 6.5 J/g after HPP in contact with oil simulant, but increased to 17.7 J/g after HPP with aqueous simulant; similarly, Δ*H*_c_ of the untreated film decreased from 9.7 to 4.9 J/g after HPP in contact with oil simulant, but increased to 15.9 J/g after HPP with aqueous simulant; % χ_c_ of the untreated film 2.4% decreased to 1.7 and 1.9% for HPP with oil simulant and HPP with aqueous simulant, respectively when subjected to 500 MPa/50 °C/15 min	--	[12]
PLA	200, 500 and 700 MPa for 5 min at 90 °C (sterilization) and 28.5 °C (pasteurization)	tap water, solid carrots, carrot puree and carrot juice	WVP of untreated PLA film at 25 and 30 °C was 1.47 × 10^−8^ and 1.40 × 10^−8^ g cm/(cm^2^ atm s) decreased to 1.22 × 10^−8^ and 1.14 × 10^−8^ g cm/(cm^2^ atm s), respectively when subjected to PLA pasteurized (carrot juice) at 700 MPa. But HP sterilization caused unacceptable embrittlement and opacification of the PLA film and thus WVP was not performed	--	--	Two T_g_ of untreated film 55.8 (associated with the amorphous external layers) and 62.45 °C (inner core of the film) showed no significant variation for pasteurization of tap water at 700 MPa and T_g_ was 55.6 and 62.25 °C, but T_g_ after sterilization of tap water at 700 MPa increased to 58.85 and 63.7 °C. Similarly, χ_c_ of the untreated film was 25.55% showed no significant variation for pasteurization of tap water at 700 MPa and 28.5 °C and it was 25.15% but increased to 29.4% after sterilization of tap water at 700 MPa and 90 °C	--	[54]
PLA and Ag (1, 5, 10, 15, and 20 wt%)	100, 200, 300, and 400 MPa for 10, 20, and 30 min	Isooctane	HPP (400 MPa for 20 min) increased the WVP of the nanocomposite film as the migration time increased and WVP was higher for the nanocomposite containing a higher content of Ag nanoparticles	HPP (400 MPa for 20 min) had no significant impact on TS of the composite film	HPP (400 MPa for 20 min) decreased the EAB with the increase in the migration time	HPP (400 MPa for 20 min) had no significant impact on T_m_ of the composite film; T_g_ of the composite films enhanced with the increase in the migration time but was independent of the nano-Ag content. T_c_ and χ_c_ increased as migration time and nano-Ag content increased	Migration Ag NPs enhanced with the increase in NPs concentration, pressure level, and migration time	[33]
PLA, polyethylene glycol (PEG), and GO nano-sheets (0, 1, and 2%)	300, 450, and 600 MPa at 25–38 °C for 15 min	--	WVP of untreated PLA-PEG-GO-2% was 1.05 ± 0.11 × 10^−14^ (kg m) (m² s Pa) increased to 1.30 ± 0.04 × 10^−14^, 1.57 ± 0.24 × 10^−14^, and 1.68 ± 0.21× 10^−14^ (kg m) (m² s Pa); similarly, OP of untreated PLA-PEG-GO-2% was 2.18 ± 0.12 × 10^−18^ (kg m) (m² s Pa) increased to 3.09 ± 0.20 × 10^−18^, 4.44 ± 0.10 × 10^−18^, and 6.54 ± 0.37 × 10^−18^ (kg m) (m² s Pa) when subjected to 300, 450, and 600 MPa, respectively	TS of untreated PLA-PEG-GO-2% was 50.80 ± 0.75 MPa decreased to 43.13 ± 6.64, 40.69 ± 0.77, and 40.14 ± 1.00 MPa when subjected to 300/450/600 MPa, respectively	EAB of untreated PLA-PEG-GO-2% was 25.31 ± 0.27% decreased to 20.32 ± 1.35, 17.98 ± 0.92, and 11.72 ± 1.35% when subjected to 300/450/600 MPa, respectively	T_g_ of the untreated PLA-PEG-GO-1% film 38.45 ± 0.89 °C, increased to 43.09 ± 0.39, 43.21 ± 0.97, and 49.35 ± 0.68 °C; Two T_m_ of untreated was film was 141.87 ± 0.89 and 149.68 ± 1.33 °C increased to 144.59 ± 1.22 and 150.40 ± 1.10, 141.66 ± 1.19 and 149.88 ± 0.87; 146.40 ± 1.24 and 150.29 ± 1.41 °C; T_c_ of the control film was 103.30 ± 0.58 °C increased to 108.45 ± 0.66, 110.10 ± 0.99, and 116.85 ± 0.98 °C; %χ_c_ of the untreated film was 26.42 ± 0.78% increased to 33.01 ± 1.13, 34.21 ± 1.20, and 35.76 ± 1.15% when subjected to 300/450/600 MPa, respectively	--	[51]
Cellulose acetate (CA)	200, 300 or 400 MPa for 5 or 10 min	--	WVTR of untreated CA film was 232.56 ± 2.29 g.m^−2^. day^−1^ significantly decreased with the increase in pressure levels as follows: 205.57 ± 4.48 and 197.84 ± 1.86 g.m^−2^. day^−1^ when exposed to 200 MPa for 5 and 10 min, respectively; 192.35 ± 1.13, and 185.78 ± 1.77 g m^−2^ day^−1^ when exposed to 300 MPa for 5 and 10 min, respectively; 182.53 ± 0.68 and 177.36 ± 2.26 g.m^−2.^ day^−1^ when exposed to 400 MPa for 5 and 10 min, respectively	TS of untreated CA film was 40.9 ± 1.2 MPa significantly decreased to 34.5 ± 1.5, and 34.9 ± 1.1 MPa when exposed to 200 MPa for 5 and 10 min, respectively; 28.9 ± 1.3 and 36.6 ± 1.3 MPa when exposed to 300 MPa for 5 and 10 min, respectively; 32.4 ± 0.7 and 32.6 ± 1.4 MPa when exposed to 400 MPa for 5 and 10 min, respectively	EAB of untreated CA film was 4.2 ± 0.1% significantly increased to 5.5 ± 0.3 and 5.5 ± 0.3% when exposed to 200 MPa for 5 and 10 min, respectively; 6.3 ± 0.4 and 5.3 ± 0.3% when exposed to 300 MPa for 5 and 10 min, respectively; 6.2 ± 0.3 and 5.8 ± 0.3% when exposed to 400 MPa for 5 and 10 min, respectively	Tg of untreated CA film was 203 °C, decreased to 197.25, and 195.33 °C when exposed to 200 MPa for 5 and 10 min, respectively; decreased to 196.88, and 196.08 °C when exposed to 300 MPa for 5 and 10 min, respectively; decreased to 196.95, and 196.95 °C when exposed to 400 MPa for 5 and 10 min, respectively. Similarly, T_m_ of untreated CA was 227.40 °C and showed minimal change in a narrow range between 227.66 to 226.59 °C when exposed to 400 to 200 MPa for 10 min	--	[56]
Co-extruded PLA	300, 450, and 600 MPa for 15 min at 25–38 °C	--	WVP of the untreated film (25-μm) was 2.05 ± 0.43) × 10^−14^ kg·m/ (m²·s·Pa) showed no significant variation in the WVP and the values were (2.02 ± 0.23) × 10^−14^, (1.95 ± 0.11) × 10^−14^, and (2.13 ± 0.10) × 10^−14^ kg·m/ (m²·s·Pa) when subjected to 300, 450, and 600 MPa; but WVP of the 75-μm showed no significant variation when subjected to HPP. OP of the untreated film (25-μm) was (6.55 ± 0.41) × 10^−18^ [kg·m/ (m²·s·Pa)] and significantly increased to (9.67 ± 0.84) × 10^−18^ [kg·m/ (m²·s·Pa)] when subjected to 600 MPa and 75-μm showed no significant variation of the OP when subjected to HPP	--	--	T_m_ of both the untreated film was 166.87 °C and HPP (600 MPa) showed no significant variation. Similarly, fusion enthalpies (ΔE) of untreated 25 and the 75-μm film was 35.10 and 34.41 J/g values changed insignificantly (34.37–38.13 J/g) after the HPP except for 25-μm film with 600 MPa (31.43 J/g)	--	[57]
PLA and TiO_2_ nanoparticles	300 MPa for 10 min	water	WVP of the untreated PLA loaded with 0, 10, and 20% TiO_2_ nanoparticles film was 5.28 ± 0.08, 4.78 ± 0.26, and 5.33 ± 0.17 (g·m)/(m^2^·s·Pa) significantly decreased to 4.81 + 0.17, 4.12 + 0.15, and 4.97 + 0.18 (g·m)/(m^2^·s·Pa), respectively 300 MPa for 10 min	TS of untreated film containing 0, 10, and 20% TiO_2_ nanoparticles film was 30.71 ± 1.18, 34.89 ± 1.19, and 32.45 ± 1.42 MPa showed no significant variation and values were 31.32 ± 0.94, 36.08 ± 1.25, and 33.72 ± 1.78 MPa, respectively when subjected to 300 MPa for 10 min	EAB of untreated film containing 0, 10, and 20% TiO_2_ nanoparticles film was 83.7 ± 5.21, 72.1 ± 6.81, and 75.2 ± 5.12% decreased to 79.1 ± 5.86, 64.2 ± 4.62, and 72.1 ± 5.67%, respectively when subjected to 300 MPa for 10 min	T_g_, T_c_, T_m_, and χ_c_ of untreated PLA film were 45.2 °C, 112.2 °C, 168.2 °C, and 14.5% changes insignificantly to 45.8 °C, 108.3 °C, 172.1 °C, and 18.3% when subjected to 300 MPa for 10 min; Similarly, T_g_, T_c_, T_m_, and χ_c_ of untreated PLA/ TiO_2_-10% film was 49.7 °C, 105.2 °C, 168.4 °C, 18.7% changes to 48.8 °C, 102.5 °C, 168.5 °C, 22.4%, respectively when subjected to 300 MPa for 10 min	--	[55]
PLA and TiO_2_ nanoparticles	300 MPa for 10 min	Ethanol Solution	WVP of HPP of PLA 4.81 ± 0.17 (g·m)/(m^2^·s·Pa) (on 0 day) showed no significant variation over the period of 30 day, 5.55 ± 0.15 (g·m)/(m^2^·s·Pa) (on 30th day); similarly for PLA- TiO_2_-20%, WVP 4.97 ± 0.18 (g·m)/(m^2^·s·Pa) (on 0 day) showed no significant variation over the period of 30 days, 5.61 ± 0.13(g·m)/ (m^2^·s·Pa).Similarly, OP of PLA film 4.02 ± 0.18 [(cm^3^/(24 h m^2^)].(cm/bar) (on 0 day) showed no significant variation over the period of 30 day, 4.77 ± 0.15 (cm^3^/(24 h.m^2^)].(cm/bar); similarly for PLA- TiO_2_-20%, OP 3.98 ± 0.21 (cm^3^/(24 h m^2^)].(cm/bar) (on 0 days) showed no significant variation over the period of 30 days, 4.81 ± 0.20 (cm^3^/(24 h m^2^)].(cm/bar)	--	--	T_g_, and T_c_, of PLA film after HPP at 300 MPa/10 min on 0 day was 46.0 °C, and 119.2 °C, increased to 59.9 °C, and 121.5 °C, respectively on the 30^th^ day; T_m_ on 0 day was 171.3 °C insignificantly deceased to 169.4 °C on 30^th^ day; χ_c_ on 0 day was 20.5% decreased to 15.2% on 30^th^ day; similarly, thermal properties after HP treatment on 0 to 30^th^ day for PLA- TiO_2_-20% was as follows: T_g_ increased from 48.8 to 62.5 °C; T_m_ value increased from 108.6 to 119.0 °C, T_c_ value 170.0 °C insignificantly changed to 170.5 °C; χ_c_ value decreased from 25.4% to 19.4%	Under HPP, migration of TiO_2_ NPs increased with the increase in the concentration of NPs, and migration time	[60]
PLA/Uvitex OB^®^	800 MPa at 20, and 90 °C for 5 min	Distilled water 3% acetic acid, 15% ethanol, Olive oil	--	--	--	--	The migration of Uvitex OB^®^ due to HPP was too low to be detected	[61]
Co-extruded PLA	450, and 600 at 26–39 °C for 15 min	water	WVP of the untreated film (1.55 ± 0.12) × 10^−14^ (kg·m/[m^2^·s·Pa]) significantly increased to (1.59 ± 0.10) × 10^−14^ and (1.62 ± 0.13) × 10^−14^ (kg·m/[m^2^·s·Pa]) when subjected to 450 and 600 MP, respectively; OP of the untreated film (6.58 ± 0.33) × 10^−18^ (kg·m/[m^2^·s·Pa]) significantly increased to (8.67 ± 0.26) × 10^−18^ and (9.16 ± 0.44) × 10^−18^ (kg·m/[m^2^·s·Pa]) when subjected to 450 and 600 MP, respectively	TS of the untreated film 36.40 ± 1.11 MPa significantly decreased to 33.04 ± 0.45 and 31.02 ±1.97 MPa when subjected to 450 and 600 MPa, respectively	EAB of the untreated film 31.92 ± 2.81% significantly decreased to 21.58 ± 2.76 and 17.54 ± 1.39% when subjected to 450 and 600 MPa, respectively	T_g_ and T_m_ of the untreated film do not show significant variation with the increase in the pressure level, however, T_c_ of the untreated film 118.08 ± 0.56 °C significantly increased to 120.34 ± 0.59 °C when subjected to 450 MPa but decreased to 117.33 ± 0.62 MPa when treated with 600 MPa; χ_c_ of the untreated film 26.35 ± 0.21% decreased to 25.80 ± 0.22 and 19.98 ± 0.19% when subjected to 450 and 600 MP, respectively	--	[58]
PLA (Biophan 121 of 40 μm)	800 MPa, 40, and 115 °C for 5 min	Distilled water, 3% acetic acid, 15% ethanol, and Olive oil	--	--	--	--	HP pasteurization at 800 MPa, 40 °C for 5 min, significantly decreased the absorption of aroma compounds, and the losses of ethyl hexanoate were up to 17%; But HP sterilization at 800 MPa, 115 °C for 5 min, significantly increased the absorption of aroma compounds and the losses of ethyl hexanoate were up to 60%	[62]
Wheat gluten, montmorillonite (MMT), Uvitex OB^®^	800 MPa at 20, 90 °C for 5 min	Distilled water, 3% acetic acid, 15% ethanol, and Olive oil	--	--	--	--	HP pasteurization (800 MPa at 20 °C for 5 min) had no impact on migration behavior, but HP sterilization (800 MPa at 20 °C for 5 min) melted the film	[63]

OTR: Oxygen transport rate, OP: Oxygen permeability; WVTR: Water vapor transport rate; WVP: Water vapor permeability; TS: Tensile strength; EAB: Elongation at break; T_g_: Glass transition temperature; T_m_: Melting temperature; T_c_: Crystallization temperature; χ_c:_ % Crystallization, PEG: Polyethylene Glycol, CIN: Cinnamon oil, GO: Graphene Oxide.

## Data Availability

Data sharing is not applicable.

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
