# Peer review of "Effect of High-Pressure Processing on the Packaging Properties of Biopolymer-Based Films: A Review"

_polymers, 2022, doi:10.3390/polym14153009_

Round 1

Reviewer 1 Report

The paper is well written and contains several interesting up-to-date data and critics.

Abstract contains too much introduction. The abstract should majorly and specifically focus on the content of the present paper. What will the reader get from this review?

L96-110 Biopolymers are mostly water sensitive/solubility (partial or fully). How can they used in HPP? Packaging in another bags?

Table 1 caption Add that they are HPP treatments of FFS.

Section 3 Are these films packaged in bags prior to HPP?

L476 This means there is PLA package for HPP available in the market?

Reviewer 2 Report

The paper seeks to introduce an approach “Effect of High-Pressure Processing on the Packaging Properties of Biopolymer-Based Films: A Review”. However, the authors should consider improving upon the quality to further highlight and emphasis. 

1.    Based on the understanding of what should be included in the abstract, consider adding one or two lines highlighting the significance of this review article.

2.    The introduction needs to be improved by relating to the mechanics of the studied materials and their mechanical characteristics. The references to be included are: 10.1007/s10853-022-06994-3, 10.1177/0021998318790093, 10.1016/j.polymertesting.2017.09.009, 10.1016/j.compstruct.2021.114698, 10.1177/0731684417727143 and 10.1002/app.46770.

3.    Consider putting space between a variable and its corresponding unit. Instead of writing 27%, write 27 %, and also instead of $9.8 billion, write $ 9.8 billion.

4.    Increase the font size of figure 2.

5.    Also, one standard of spelling should be adopted. It should be either figure or fig but not both. Consider adopting one style. In paragraph 2 of your introduction, you wrote “fig.” and in the last paragraph of the same introduction, it appeared as “figure”.

6.    The magnification footers in figures 3 and 5 could not be read. Manually indicate all inside the image.

7.    You have done a comprehensive work but give us a future perspective that paves way for a research gap for other researchers of interest to follow.

Consider providing a scheme on how this whole review is being addressed.
